# Non-Linear Regression Models with Vibration Amplitude Optimization Algorithms in a Microturbine

**DOI:** 10.3390/s22010130

**Published:** 2021-12-25

**Authors:** Omar Rodríguez-Abreo, Juvenal Rodríguez-Reséndiz, L. A. Montoya-Santiyanes, José Manuel Álvarez-Alvarado

**Affiliations:** 1Industrial Technologies Division, Universidad Politécnica de Querétaro, El Marques 76240, Mexico; omar.rodriguez@upq.edu.mx (O.R.-A.); luis.montoya@upq.edu.mx (L.A.M.-S.); 2Red de Investigación OAC Optimización, Automatización y Control, El Marques 76240, Mexico; 3Facultad de Ingeniería, Universidad Autónoma de Querétaro, Querétaro 76010, Mexico; jmalvarez@uaq.mx

**Keywords:** machine diagnosis, mechanical sensors, vibration, non-linear model, grey wolf optimizer (GWO), metaheuristics algorithms

## Abstract

Machinery condition monitoring and failure analysis is an engineering problem to pay attention to among all those being studied. Excessive vibration in a rotating system can damage the system and cannot be ignored. One option to prevent vibrations in a system is through preparation for them with a model. The accuracy of the model depends mainly on the type of model and the fitting that is attained. The non-linear model parameters can be complex to fit. Therefore, artificial intelligence is an option for performing this tuning. Within evolutionary computation, there are many optimization and tuning algorithms, the best known being genetic algorithms, but they contain many specific parameters. That is why algorithms such as the gray wolf optimizer (GWO) are alternatives for this tuning. There is a small number of mechanical applications in which the GWO algorithm has been implemented. Therefore, the GWO algorithm was used to fit non-linear regression models for vibration amplitude measurements in the radial direction in relation to the rotational frequency in a gas microturbine without considering temperature effects. RMSE and $R^{2}$ were used as evaluation criteria. The results showed good agreement concerning the statistical analysis. The 2nd and 4th-order models, and the Gaussian and sinusoidal models, improved the fit. All models evaluated predicted the data with a high coefficient of determination (85–93%); the RMSE was between 0.19 and 0.22 for the worst proposed model. The proposed methodology can be used to optimize the estimated models with statistical tools.

## 1. Introduction

Nowadays, the modernization of industry is encouraging researchers to deploy techniques to increase energy efficiency [1]. This has made it possible to integrate distributed energy resources (DER) to reduce fossil fuel-based energy generation by using renewable energies [2]. DER technologies are mainly designed for small-scale energy generation systems to provide a local solution by enacting smart management of the available energy and storage systems [3,4].

Microturbines play an important role in distributed generation applications which are small-scale due to their great performance in terms of efficiency and small size [1,5]. The control of a microturbine is a challenge due to the unbalanced forces, internal self-excitation, external excitation and complex work environment, to name a few. In particular, excessive vibrations in rotating systems cannot be ignored [6]. In this context, the study of vibrations can improve the control of a microturbine to obtain high efficiency in these systems [7,8].

Conventional models for the study of vibrations of a microturbine in different applications focus only in the dynamic model [9,10,11]. However, obtaining better energy management requires optimizing the efficiency of DER systems. Several investigations have implemented algorithms to optimize significant parameters in engineering problems. Metaheuristic algorithms perform well in various engineering problems. For example, the have been applied in gear system design, cam design, wind turbine blade design, aeronautical equipment and combustion systems [12]; and in engineering areas such as robotics, data mining, security and many others [13].

To handle vibrations, metaheuristics algorithms can be used to increase the efficiency by maximizing the amplitude of the defect frequency [14]. Another work in [15], found that by optimizing the parametric shape of a cam for any mechanism by genetic algorithm (GA), one can minimize vibration and improve dynamic performance. Li et al. [16] implemented a GA to find the vibration modes of different composite laminated elements and showed that computational cost could be reduced compared with finite element analysis (FEA). The parameters were the unbalance force, axial position and phase. If there is no difference between the amplitude and phase of the experimental and calculated signals, the model represents the physical system well.

It has also been shown that many authors develop analyses using Design of Experiments (DOE) and optimize regression polynomials using metaheuristic algorithms [17,18]. The gray wolf optimizer (GWO) algorithm is among the new generation metaheuristic algorithms and is adaptable to multiple problems [19]. The GWO is considered one of the fastest-growing algorithms among the swarm intelligence (SI) algorithms. The algorithm is inspired by gray wolves in nature, seeking the optimal way to hunt their prey [20]. In this sense, the parameters that best fit a model to the desired value can be found. Heuristic optimization methods focus on a near-optimal solution to the problem but do not focus on the best solution. The number of publications using the GWO algorithm in the last decade, including machinery analysis applications, remains small [21]. The GWO algorithm’s performance has been demonstrated in a study on the tension/compression of a spring in a welded beam, and in the design of pressure vessels; specifically it works in unknown search spaces and finds the optimal parameters quickly [22].

All rotating machines maintain a residual unbalance that results in the rotational or synchronous frequency. The frequencies of vibration associated with the most common faults in rotating machinery, such as imbalance, misalignment, backlash and bearing orbits, are multiples or percentages of the rotation frequency. The unbalance increases the amplitude of the rotational frequency mainly in the radial direction [23]. To conclude, we may say that there are very few mechanical applications where the GWO algorithm has been implemented, and even fewer in vibration analysis [24].

The literature has proposed different techniques to prevent vibrations by applying metaheuristic solutions in the dynamic model, but it is still challenging to get a response fast enough in vibration analysis. This article presents a GWO algorithm to improve the performance of a microturbine. It is used to adjust non-linear regression models to measure the amplitude of vibration in the radial direction in relation to the rotational frequency in a gas microturbine without considering the effects of temperature. The novelty of the work is the analysis of six different types of non-linear model, all of which were highly adjusted thanks to the GWO. The adjustments were carried out using data obtained experimentally with high variability, making the regression consider the standard errors expected in the measurements. The results show a mean RMSE of 0.20 for all models and regression coefficients of 85% for the worst model and 92% for the best.

The rest of the work is organized as follows. In Section 2, we report the methodology for data collection, the non-linear models and the use of the GWO as a parameter adjuster. Section 3 describes the results obtained for each coefficient of the non-linear models and their performances as estimators of vibration. Section 4 briefly discusses the results, and finally, Section 5 shows the conclusions of this investigation.

## 2. Materials and Methods

This section describes the process used to estimate the parameters of the non-linear models, for which the first step is the collection of experimental vibration data. For the data acquisition, the gas microturbine shown in Figure 1 was used. It has a compressor made of PLA 3D printing material, a 304 stainless steel turbine and HSS steel shaft. A piezoelectric accelerometer (PCB Piezotronics, Model: 333B30, sensitivity: 98.2 ${}^{\mathrm{mV}}{\;/\;}_{g}$) was used to measure the acceleration response. Table 1 summarizes the technical data of the microturbine used in the experimentation. The data acquisition system consisted of the National Instruments NI-9234 module and a cDAQ-9174 chassis. The signal processing was performed using LabVIEW. The accelerometer was positioned close to the bearing on the side of the compressor wheel—see Figure 1—avoiding contact with the support straps of the system. In this position, it would be away from the highest temperature zone if tests involving combustion are required. It is worth mentioning that the scope of this study did not include combustion tests.

Most rotor or shaft failures occur at speeds coinciding with the rotational frequency or multiples of the latter, harmonics or subharmonics [25]. It was decided to use a constant air supply source at the inlet of the microturbine because the effect of temperature will not be considered, and therefore, no combustion causes acceleration. The maximum speed reached in this way was 127 Hz. Consequently, it was decided to use the nominal rotational frequencies of 27, 76 and 127 Hz, equivalent to 1620, 4560 and 7620 rpm, and thus have different factor levels. A total of 10 reps were performed for each frequency. The established frequencies were similar to those used in Kumbhar et al. [26], although they mainly focused on bearings.

Figure 2 shows, on a linear-linear scale, the average vibration amplitude response for each nominal frequency.

The amplitude values in Figure 2, as specified above, are the averages. This tends to graphically smooth the value of the peak that each repeat had. However, due to the variability in the measurements, the real average frequency values consider the peak of each repetition. The average amplitude, considering the peaks of each repetition and its variations, are shown in Table 2.

A series of non-linear models were selected to adjust the vibration amplitude measurements and evaluate the best score through the root mean square error (RMSE) and the correlation coefficient $R^{2}$. A total of 10 repetitions were performed for each model. The proposed models consist of 2nd, 3rd and 4th-order polynomials; and exponential, Gaussian and sinusoidal models.

The equations are summarized from (Equation 1) to (Equation 6), respectively, where the frequency is handled as *x* and the letters *a*, *b*, *c*, *d* and *e* are the unknown coefficients.
(1)$$a{\left(x\right)}^{2}+b\left(x\right)+c$$
(2)$$a{\left(x\right)}^{3}+b{\left(x\right)}^{2}+c\left(x\right)+d$$
(3)$$a{\left(x\right)}^{4}+b{\left(x\right)}^{3}+c{\left(x\right)}^{2}+d\left(x\right)+e$$
(4)$$a{\left(\exp\right)}^{b(x)}+c{\left(\exp\right)}^{d(x)}$$
(5)$$a{\left(\exp\right)}^{-1\frac{{\left(x-b\right)}^{2}}{2{\left(c\right)}^{2}}}$$
(6)a(sin(b(x)+c))

The objective was to compare the results with the statistically estimated model in [27] regardless of whether or not there are non-significant terms in the latter.

For the adjusting, the GWO was used. The original diagram of the GWO algorithm can be seen in Figure 3, and its full description can be found in [22]. Within the metaheuristic algorithms, the GA is the most widely used. However, it presents the significant disadvantage of having multiple specific parameters to adjust. In contrast, the wolf algorithm, in addition to the advantages indicated in the introduction, does not have any specific extrinsic parameter, which becomes one of the options for calculating the coefficients.

All metaheuristic algorithms are susceptible to local optima, which is why cross-referencing with different initial values was used to verify that the algorithm was not remaining fixed in a local optimum. Through the GWO algorithm, the coefficients with the lowest RMSE in a fixed number of iterations were calculated. All the models were adjusted with the same search parameters, which are summarized in Table 3. The general search parameters allowed us to adjust how the GWO finds the solution. The number of agents and iterations remained fixed for all models. A large number of agents helps in the search and avoids local optimum, but increases the computational cost. The upper and lower search limits determine the search limits of the parameters of each model, and these must be adapted to the type of model.

It can be observed that the search parameters from Table 3 that the search was carried out with a fixed number of iterations and with the same number of search agents, while slightly varying the search limits in the parameters according to the part they represent in each non-linear model. It is important to note that as stated in the no free lunch theorem, there is no algorithm superior to any other if they are averaged in all cases [28]. This is why any metaheuristic algorithm without specific parameters can solve the task.

## 3. Results

Table 4 displays the values of the coefficients calculated for each model. These correspond to the samples that resulted in the best scores. The results for the second, third and fourth-order polynomial models clearly show that the coefficients of the higher-order terms than the quadratic are much less significant, which is why these coefficients’ values are much lower than those of the terms quadratic, linear and independent. The exponential results model indicates similar exponent values. However, a different sign is shown in the coefficients to allow the change in slope direction with this model. This model shows a slowly rising slope and a slow descending slope, unlike a polynomial. This is caused by the difference in the magnitude of the coefficients of both exponentials. For the Gaussian function, the three parameters calculated were coefficient a, which is the value of the highest point of the bell; b is the position of the center of the bell; and c is the standard deviation. Thus, parameter a was determined by the magnitude of the measurements, and the average frequency of the measurements determined parameter b. Finally, parameter c reflects the variability of the model. For the sinusoidal function, parameter a is the amplitude, which is why in magnitude, it is virtually the same as the parameter an of the Gaussian model. The angular velocity is coefficient b, and the phase is coefficient c. The regression of each model on the data is depicted in Figure 4, Figure 5, Figure 6, Figure 7, Figure 8 and Figure 9 as follows:Figure 4a corresponds to the quadratic regression, and Figure 4b shows how the RMSE is reduced as the number of iterations is increased for the 2nd-order model.Figure 5a corresponds to the cubic regression, and Figure 5b presents how the RMSE is reduced as the number of iterations is increased for the 3rd-order model.Figure 6a corresponds to the quartic regression, and Figure 6b shows how the RMSE is reduced as the number of iterations is increased for the 4th-order model.Figure 7a corresponds to the exponential regression, and Figure 7b displays how the RMSE is reduced as the number of iterations is increased for the exponential model.Figure 8a corresponds to the Gaussian regression, and Figure 8b depicts how the RMSE is reduced as the number of iterations is increased for the Gaussian model.Figure 9a corresponds to the trigonometric regression, and Figure 9b shows how the RMSE is reduced as the number of iterations is increased for the sinusoidal model.

The number of iterations in which the minimum RMSE was reached varied in each repetition for each model. For the 2nd-order model, it ranged from 322 to 474; for the 3rd-order model, it ranged from 430 to 500; for the 4th-order model, it ranged from 458 to 498; for the exponential model, it ranged from 449 to 500; for the Gaussian model, it ranged from 438 to 496; and for the sinusoidal model, it ranged from 338 to 485.

The results of the RMSE and $R^{2}$ for all models are shown in Table 5. The computational performance and mean bias error (MBE) are presented in Table 6. The standard deviation is included in both due to repetitions being performed. The figures of the models show symmetric behavior for the quadratic, gaussian and sinusoidal models. In contrast, the cubic, quartic and exponential models show asymmetric behavior with different rising and falling slopes. This effect can be observed more clearly in the Exponential model (Figure 7a). The values of the parameters were adjusted based on the RMSE. However, these do not reflect the data adjustment or the variability of the models. Therefore, the $R^{2}$ and the standard deviation of each model are in Table 4. It is observed that practically all the models share the same RMSE, the exception being the exponential model, which presents the most significant error. However, the third and fourth-order polynomial models show the lowest standard deviation, and the cubic model the worst $R^{2}$. In general terms, the second-order, Gaussian, and sinusoidal models offer the best behavior—practically the same in RMSE, $R^{2}$ and standard deviation.

If we use all the proposed models to predict the amplitudes of vibration at the specific average frequencies from Table 2, the relative errors are those shown in the following Table 7. This was done for a comparison with the results of the quadratic model reduced in [27].

## 4. Discussion

All the models predicted the data with almost 93% accuracy, except for the 3rd-order model. This is very close to what was published in [26], where the vibration amplitude variations were predicted with 95% accuracy.

The 4th-order model is the one that required the longest computation time. However, it is the one with the best coefficient of determination. Additionally, this model produced the lowest RMSE value of all. In a previous study [27], the 2nd-order model was fitted using one-way ANOVA. If we take the $R^{2}$ of the statistical 2nd-order model in the study referenced previously, the 2nd-order model had a 0.33% higher $R^{2}$. The 3rd-order model had a 7.65% lower $R^{2}$, the 4th-order model had a 0.34% higher $R^{2}$, the exponential model had a 0.78% lower $R^{2}$, the Gaussian model had a 0.33% higher $R^{2}$ and the sinusoidal model had a 0.34% higher $R^{2}$. Thus, taking the 2nd-order model of this paper as a reference, the 3rd-order model had a 0.1% higher RMSE, the 4th-order model had a 0.13% lower RMSE, the exponential model had a 14.52% higher RMSE, the Gaussian had a 0.01% lower RMSE and the sinusoidal model had a 0.09% lower RMSE.

Therefore, the 4th-order model had the best score regarding the evaluation criteria, although it must be considered that the computational time was 200% greater than that of the 2nd-order model. The 2nd-order model had the least computation time. In this case, the Gaussian and sinusoidal models can also be selected because they had very similar $R^{2}$ values and only minor reductions in RMSE, but computation times closer to the low end.

Although the 4th-order and exponential models did not produce MBE of the great magnitude, those of the other models were very close to zero.

The models had the following average relative errors: 0.038% for the 2nd-order model, 0.065% for the 3rd-order, 0.157% for the 4th-order, 0.911% for the exponential, 0.023% for the Gaussian and 0.011% the for sinusoidal. The sign for a value of Table 7 means whether the model overestimated or underestimated the amplitude value.

In general, the models produced average relative errors lower than the reduced second-order statistical model in the evaluated frequency spectrum, which was 5%.

By varying the rotational frequency, Figure 2, a "beat vibration" effect should be appreciated, very similar to that shown in [29]. The increases in the amplitude values of vibration in the second speed may be because this frequency is close to the natural frequency of the rotor. However, the measurements of the natural frequencies are outside the scope of this work.

In addition, the models follow a pattern similar to that shown in [30], where the effect of the spindle speed on the amplitude of vibration is significant in any direction.

Likewise, the regression models are similarly adjusted to the data shown in [31], where they experimentally analyzed the behavior of cracked rotor in the presence of torsional vibrations. Subsequently, it was determined that the amplitude of lateral vibration at the fundamental frequency increases with the rotation speed, regardless of rotor crack condition (transverse and 45°) or normal condition, with or without torsional excitation.

A methodology to develop an intelligent monitoring system for machining processes was developed by [18]. As well as in this methodology, metaheuristic algorithms can be implemented in conjunction with statistical tools such as the Design of Experiments to improve the modeling of mechanical processes.

The results evidently show that metaheuristic algorithms serve to optimize regression models satisfactorily. The absence of implementations in mechanical systems over the last few years may have been due to few comparative and standardized results in the literature, the fact that they have several parameters that take a long time to determine their values or that there may be stagnation in local optima [24].

## 5. Conclusions

In this study, the GWO algorithm was implemented to adjust non-linear regression models to measurements of the amplitude of vibrations in relation to the rotational frequency in a gas microturbine.

The evaluated models can be used to predict vibrations with a high coefficient of determination by evaluating a specific factor, not necessarily rotational frequency only, and considering the appropriate operating parameters. Additionally, the methodology proposed in this paper can be used to optimize models that were estimated with statistical tools, such as DOE, to improve the modeling of mechanical processes.

Although the search for the parameters of each mathematical model may take more or less time, the selection of the boundary parameters for the algorithm remains simple. This facilitates its implementation in other models when the measurement of other variables is considered. Making adjustments to non-linear models considering more experimental points at higher frequencies is beyond the scope of this work, since it could happen that the determination of these models is not appropriate if the trends change drastically.

For greater precision and a better fit, a more significant number of measurement points is required. If it is possible to increase the number of measurements at each point, this allows greater reliability to each of the models. The data acquisition system is limited; consequently, the noise inherent to the measurement is one more factor to consider. It is recommended to improve the hardware’s precision (if possible) to work with less variability in the measurements.

Future work aims to analyze the models presented in this work to find response surfaces where the temperature is considered as an additional variable.

## Figures and Tables

**Figure 1 sensors-22-00130-f001:**
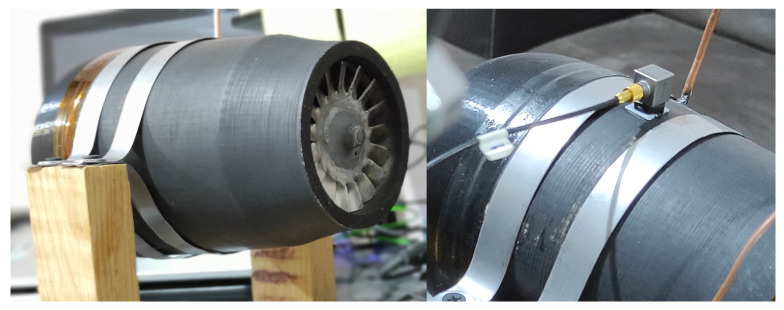
Test bench of the used microturbine.

**Figure 2 sensors-22-00130-f002:**
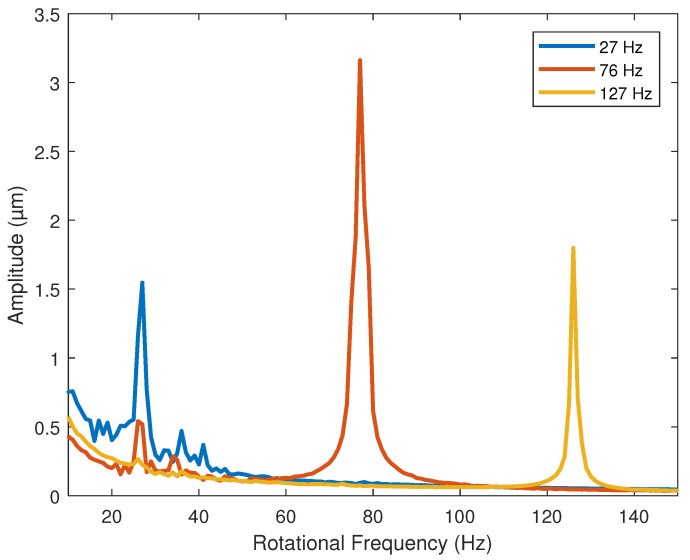
Amplitude of vibration with respect to each nominal frequency level.

**Figure 3 sensors-22-00130-f003:**
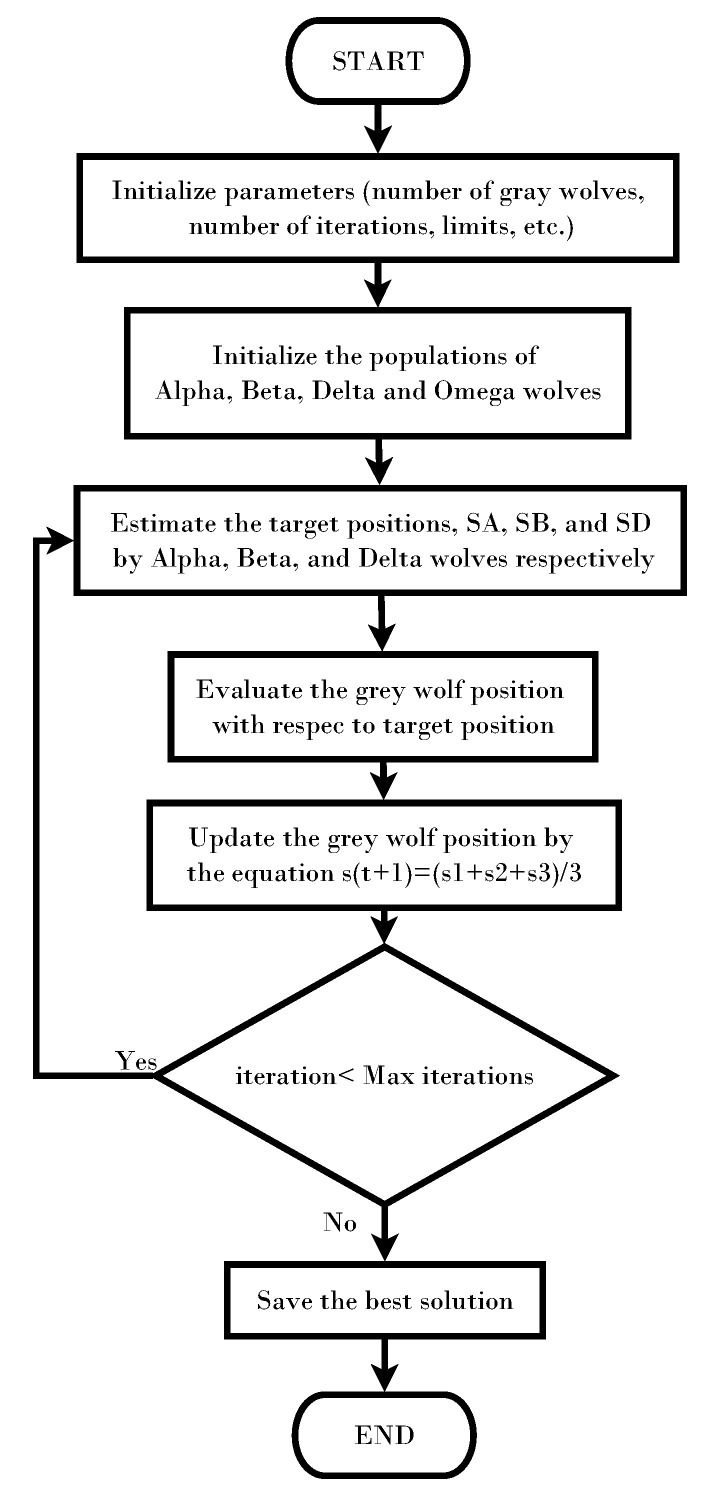
Flow chart for the GWO-based optimization.

**Figure 4 sensors-22-00130-f004:**
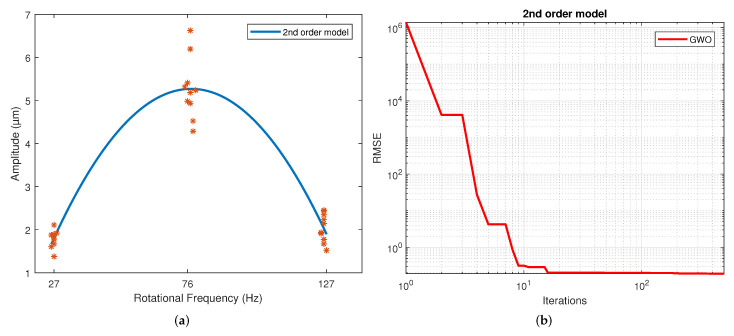
Results from the GWO using a 2nd-order model: (**a**) Regression of the 2nd-order model. (**b**) Number of iterations in which the RMSE converged with the 2nd-order model.

**Figure 5 sensors-22-00130-f005:**
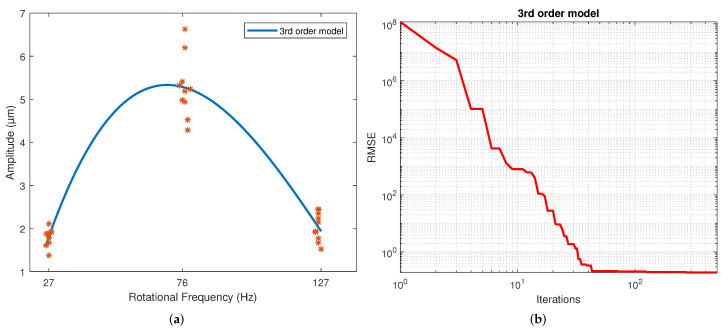
Results from the GWO using a 3rd-order model: (**a**) Regression of the 3rd-order model. (**b**) Number of iterations in which the RMSE converged with the 3rd-order model.

**Figure 6 sensors-22-00130-f006:**
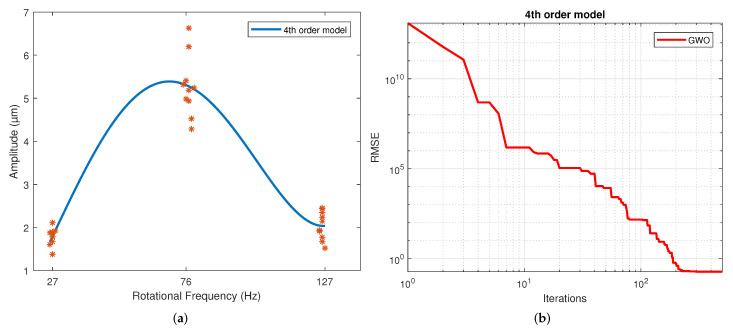
Results from the GWO using a 4th-order model: (**a**) Regression of the 4th-order model. (**b**) Number of iterations in which the RMSE converged with the 4th-order model.

**Figure 7 sensors-22-00130-f007:**
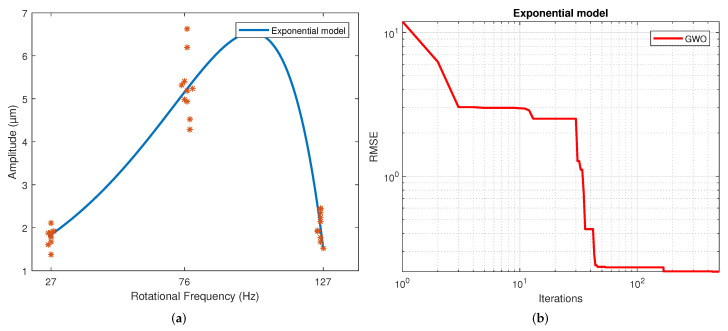
Results from the GWO using an exponential model: (**a**) Regression of the exponential model. (**b**) Number of iterations in which the RMSE converged with the exponential model.

**Figure 8 sensors-22-00130-f008:**
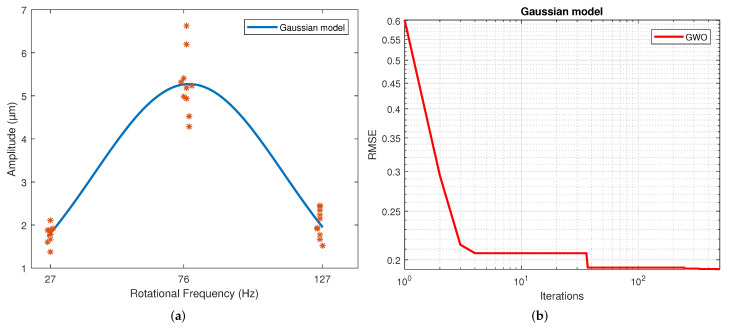
Results from the GWO using an gaussian model: (**a**) Regression of the gaussian model. (**b**) Number of iterations in which the RMSE converged with the gaussian model.

**Figure 9 sensors-22-00130-f009:**
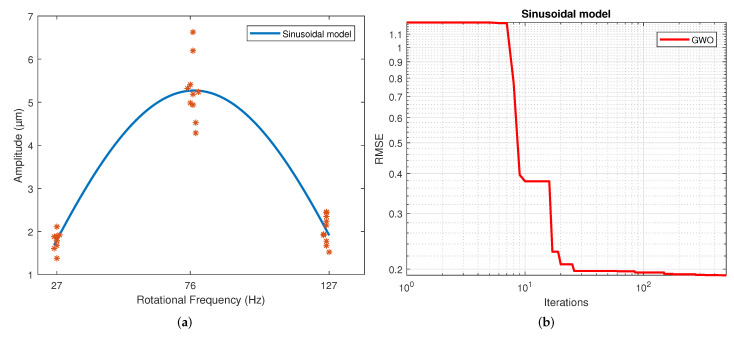
Results from the GWO using a sinusoidal model: (**a**) Regression of the sinusoidal model. (**b**) Number of iterations in which the RMSE converged with the sinusoidal model.

**Table 1 sensors-22-00130-t001:** Technical data of the microturbine.

Parameter	Description
Fuel	Butane/propane gas with maximum pressure of 3.5 kg/cm^2^
Turbine blades outer/inner diameter	68.6/40.5 mm
Compressor wheel outer/inner diameter	64.5/32.8 mm
Turbine wheel diameter	70 mm
Burner hole spacing	10 mm
Number of gas outlet holes	16

**Table 2 sensors-22-00130-t002:** Peak average frequency and amplitude values with their standard deviations.

Avg. Frequency (Hz)	Standrad Deviations	Avg. Amplitude (µm)	Standard Deviations
26.9	0.5676	1.7891	0.2009
77	1.1547	5.2697	0.7028
125.9	0.5676	2.0426	0.3287

**Table 3 sensors-22-00130-t003:** Search parameters for the model adjustment.

Parameter	2nd Order	3rd Order	4th Order	Exponential	Gaussian	Sinusoidal
SearchAgent	300	300	300	300	300	300
Iterations	500	500	500	500	500	500
Dimension	3	4	5	4	3	3
LowerBoundary	[−5 −5 −5]	[−5 −5 −5 −5]	[−5 −5 −5 −5 −5]	[−2 −5 −2 −5]	[0 0 0]	[−10 −5 −5]
UpperBoundary	[5 5 5]	[5 5 5 5]	[5 5 5 5 5]	[2 5 2 5]	[10 100 100]	[10 5 5]

**Table 4 sensors-22-00130-t004:** Resulting coefficients for the algorithm for all models.

Model	Coefficients				
	a	b	c	d	e
2nd order	−0.00136	0.21149	−2.90663	·	·
3rd order	8.01950$\;\times \;10^{-06}$	−0.00321	0.33308	−5	·
4th order	2.85$\;\times \;10^{-07}$	−7.46$\;\times \;10^{-05}$	0.00502	0.00150	−0.58955
Exponential	−0.37629	0.03966	1.30276	0.03008	·
Gaussian	5.27487	77.94529	34.75833	·	·
Sinusoidal	5.27116	0.02421	0.30448	·	·

**Table 5 sensors-22-00130-t005:** Evaluation criteria comparison for the models, including standard deviation.

Model	RMSE	$R^{2}$	Model	RMSE	$R^{2}$
2nd order	$0.19155\pm 8.0\;\times \;10^{-07}$	$0.92913\pm 2.9\;\times \;10^{-07}$	Exponential	0.21936±0.00343	0.91884±0.00126
3rd order	0.19175±0.00206	0.85524±0.23333	Gaussian	$0.19153\pm 8.1\;\times \;10^{-07}$	$0.92914\pm 3.0\;\times \;10^{-07}$
4th order	0.19129±0.00272	0.92922±0.00100	Sinusoidal	$0.19137\pm 2.3\;\times \;10^{-06}$	$0.92919\pm 8.5\;\times \;10^{-07}$

**Table 6 sensors-22-00130-t006:** Computational performance and MBE with deviations.

Model	Time (s)	MBE
2nd order	0.86225±0.01958	$1.99\;\times \;10^{-06}\pm 0.00045$
3rd order	1.69912±0.04212	$7.84\;\times \;10^{-06}\pm 0.00085$
4th order	2.61961±0.13905	0.00016±0.00361
Exponential	1.27736±0.44059	0.00059±0.00334
Gaussian	0.95463±0.02762	$4.35\;\times \;10^{-06}\pm 0.00059$
Sinusoidal	0.97166±0.03725	$7.20\;\times \;10^{-06}\pm 0.00085$

**Table 7 sensors-22-00130-t007:** Relative amplitude errors using average frequencies.

Model	26.9 Hz	77 Hz	125.9
2nd order [27]	−0.53%	0.03%	−12.88%
2nd order	0.163%	0.010%	−0.059%
3rd order	0.204%	0.004%	−0.012%
4th order	0.539%	0.010%	−0.078%
Exponential	2.774%	−0.675%	0.633%
Gaussian	0.271%	0.060%	−0.263%
Sinusoidal	0.125%	0.035%	−0.127%

## Data Availability

The data presented in this study are available on request from the corresponding author.

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
