# Peer review of "Non-Linear Regression Models with Vibration Amplitude Optimization Algorithms in a Microturbine"

_sensors, 2021, doi:10.3390/s22010130_

Round 1

Reviewer 1 Report

Excessive vibrations in rotating systems can accelerate machine failure and one option to prevent vibrations is the choice of an adequate type of the model and the fitting. The nonlinear model parameters can be complex enough to fit. Genetic algorithms are the best-known optimization and tuning algorithms, but as they include many specific parameters, the GWO algorithm was used to fit nonlinear regression models for vibration amplitude measurements relative to the rotational frequency in a gas microturbine. Very many recent works study vibrations and their model as well as their control. However, this article analyzes six types of nonlinear models with a high degree of adjustment thanks to the GWO algorithm. The evaluated models can be used to predict vibrations with a high degree of accuracy. The methodology proposed can be used to optimize models estimated with statistical tools to enhance the modeling of mechanical processes.

The authors may extend their analysis with additional methods (other AI methods and machine learning methods) which can be verified the presented results.

The work is original and contains new results that  advance the research field. The article may be accepted for publication.

Author Response

Reviewer 1

Excessive vibrations in rotating systems can accelerate machine failure and one option to prevent vibrations is the choice of an adequate type of the model and the fitting. The nonlinear model parameters can be complex enough to fit. Genetic algorithms are the best-known optimization and tuning algorithms, but as they include many specific parameters, the GWO algorithm was used to fit nonlinear regression models for vibration amplitude measurements relative to the rotational frequency in a gas microturbine. Very many recent works study vibrations and their model as well as their control. However, this article analyzes six types of nonlinear models with a high degree of adjustment thanks to the GWO algorithm. The evaluated models can be used to predict vibrations with a high degree of accuracy. The methodology proposed can be used to optimize models estimated with statistical tools to enhance the modeling of mechanical processes.

The authors may extend their analysis with additional methods (other AI methods and machine learning methods) which can be verified the presented results.

The work is original and contains new results that advance the research field. The article may be accepted for publication.

Reply:

First of all, we really appreciate your time and thank you very much for the recommendations to improve our manuscript.

  • The objective of the manuscript is not to demonstrate that using the GWO algorithm is better than using another, simply to emphasize the little use of the GWO in this type of application, as mentioned in some references of the manuscript, as well as the simplicity in the selection of parameters of the search.
  • Metaheuristic algorithms are attractive for their simplicity. In particular, the GWO does not use specific parameters. For example, the genetic algorithm was not used because it uses multiple specific parameters such as mutation and biological pressure, which require other parameters or trial and error to find them. Therefore, it does not make sense to add more unknowns by using a wide-use one.

Reviewer 2 Report

In this paper, the GWO algorithm was implemented to adjust non-linear regression models to measure the amplitude of vibration in relation to the rotational frequency in a gas microturbine. The reviewer has the following questions and the explanation need to be added to the paper:
1. What is the limitation of the GWO algorithm, and why the results are not compared with other classical algorithm?

2. Conclusion should be rewritten, to present achievements and more general conclusions. 

3. A lack of clear explanation for the advantage of the GWO algorithm.

4. Why there are few use of this method in mechanical system, and is this related to the characteristics of the system.

Author Response

In this paper, the GWO algorithm was implemented to adjust non-linear regression models to measure the amplitude of vibration in relation to the rotational frequency in a gas microturbine. The reviewer has the following questions and the explanation need to be added to the paper:

  1. What is the limitation of the GWO algorithm, and why the results are not compared with other classical algorithm?
  2. Conclusion should be rewritten, to present achievements and more general conclusions. 
  3. A lack of clear explanation for the advantage of the GWO algorithm.
  4. Why there are few uses of this method in mechanical system, and is this related to the characteristics of the system.

Reply:

First of all, we really appreciate your time and thank you very much for the recommendations to improve our manuscript.

  • Question 1: Related to other classical algorithm,

Line 96:

In previous authors work, the variation in the vibration amplitude measurements was studied in a microturbine utilizing statistical analysis using one-factor ANOVA and a quadratic regression model [27].

Line 146:

The objective is to compare the results with the statistically estimated model in [27] regardless of whether or not there are non-significant terms in the latter.

Line 192:

This is in order to compare with the results of the quadratic model reduced in [27].

It is important to note that as stated in the No Free Lunch theorem, there is no algorithm superior to any other if they are averaged in all cases [28]. This is why any metaheuristic algorithm without specific parameters can solve the task.

  • Question 2 and 3: Conclusions and advantages of GWO,

The conclusions have been modified with the intention of being more general and highlighting the advantages of the algorithm:

In this paper, the GWO algorithm was implemented to adjust non-linear regression models to measurements of the amplitude of vibration in relation to the rotational frequency in a gas microturbine.

The evaluated models can be used to predict vibrations with a high coefficient of determination by evaluating a specific factor, not necessarily rotational frequency only, and considering the appropriate operating parameters. Also, the methodology proposed in this paper can be used to optimize models that were estimated with statistical tools, such as DOE, in order to improve the modeling of mechanical processes.

Although the search for the parameters of each mathematical model may take more or less time, the selection of the boundary parameters for the algorithm remains simple. This facilitates its implementation in other models when the measurement of other variables is considered. The adjustment with the non-linear models presented considering more experimental points at higher frequencies is beyond the scope of this work, since it could happen that the determination of these models is not appropriate if the trend changes drastically.

For greater precision and a better fit, a more significant number of measurement points is required. If it is possible to increase the number of measurements at each point, this allows greater reliability to each of the models. The data acquisition system is limited; consequently, the noise inherent to the measurement is one more factor to consider. It is ideally recommended to improve the hardware precision to analyze with less variability in the measurements.

Future work aims to perform the analysis of the models presented in this work to find response surfaces where temperature is considered as an additional variable.

  • Question 4: Why there are few use of this method in mechanical system,

The following has been added to the paper in the discussion section:

The results evidently show that metaheuristic algorithms serve to optimize regression models satisfactorily. The absence of implementations in mechanical systems over the last few years may be due to few comparative and standardized results in the literature, to the fact that they have several parameters that take a long time to determine their values, or that there may be stagnation in local optimum [24].

Reviewer 3 Report

The work deals with Grey Wolf Optimizer (GWO) to fit six different nonlinear regression models for vibration amplitude measurements in the radial direction in relation to the rotational frequency in a gas microturbine. The authors compared the results with those obtained from their previous statistical analysis. The major contributions of the paper have significant merits to understand the functionality of the optimization algorithms in a microturbine. The overall presentation of the manuscript should be improved sufficiently and a few major concerns have been observed which should be addressed before it is accepted for publication. The manuscript is found with a lot of grammatical and typographical errors. 

The authors are suggested to go through the manuscript thoroughly and get it to proofread for grammatical and typographical errors.

The “Introduction” section is not at all impressive. This section should be entirely restructured instead of mentioning directly what has been done by the authors in the literature. A detailed study is required to find out the motive and novelty of the present work by doing an extensive comparison of the presented literature in the manuscript.

The authors should include more details about the experimental setup with its procedure for obtaining the experimental results in the “Materials and Methods” section for better readability. A clear and detailed justification should be mentioned for considering the GWO algorithm along with a comparative study with other algorithms. More details are required regarding the search parameters in Table 2.

The authors have presented quantitative analyses in the “Results” section, but, a detailed explanation is required for each model. There is a significant lack in explaining the plots and the results presented in Table 3-4. For e.g., in Table 3 why the values of coefficients ‘b’ and ‘c’ are very high compare to the values of the coefficients for other models. A detailed significance of the results obtained from the analyses will provide a clear idea about performing the work to the readers along with the improvements.

Overall, after a clear presentation of the results and restructured the manuscript, it has good potential for publication.

Author Response

The work deals with Grey Wolf Optimizer (GWO) to fit six different nonlinear regression models for vibration amplitude measurements in the radial direction in relation to the rotational frequency in a gas microturbine. The authors compared the results with those obtained from their previous statistical analysis. The major contributions of the paper have significant merits to understand the functionality of the optimization algorithms in a microturbine. The overall presentation of the manuscript should be improved sufficiently and a few major concerns have been observed which should be addressed before it is accepted for publication.

First of all, we really appreciate your time and thank you very much for the recommendations to improve our manuscript.

The manuscript is found with a lot of grammatical and typographical errors. The authors are suggested to go through the manuscript thoroughly and get it to proofread for grammatical and typographical errors.

Thanks for the observation. The grammar and spelling of the entire document was checked by a native speaker.

The “Introduction” section is not at all impressive. This section should be entirely restructured instead of mentioning directly what has been done by the authors in the literature. A detailed study is required to find out the motive and novelty of the present work by doing an extensive comparison of the presented literature in the manuscript.

The Introduction section was improved for a better comprehension. The motivation and relevance of the work is added in line 79, and the motive of the work was improved. 

The authors should include more details about the experimental setup with its procedure for obtaining the experimental results in the “Materials and Methods” section for better readability.

Thanks for the comment. The information of the hardware used for the experimentation as well as the hardware y software for the acquisition are described and expanded in line 90.

A clear and detailed justification should be mentioned for considering the GWO algorithm along with a comparative study with other algorithms.

Thanks for the comment. The GWO algorithm is a metaheuristic algorithm, metaheuristic algorithms have the general advantage over heuristics is their simplicity and easy implementation. Due to the high number of models that were fitted in this work, metaheuristic algorithms become a natural option for this work. Within the metaheuristic algorithms, the GWO is  one of the algorithms that does not require specific parameters, unlike genetic algorithms, which is the algorithm used in a general way. Finally, it should be considered that according to the "No Free Lunch" theorem, no optimization algorithm is better than another. Therefore, it is not intended to demonstrate the superiority of the GWO, but rather it was used for its relationship between simplicity and results. 

This is explained to the reader on lines 53 and 127.

 More details are required regarding the search parameters in Table 2.

Thanks for the comment. As mentioned in the previous point, the GWO is an algorithm that only requires specific search parameters. The general search parameters allow us to adjust the way in which the GWO finds the solution. The number of agents and the number of iterations remains fixed for all models, remembering that a large number of agents helps in the search and avoids local optimum but increases the computational cost. The upper and lower search limits allow you to determine the search limits of the parameters of each model and these must be adapted to the type of model.

This information was explained to the reader on line 138.

The authors have presented quantitative analyses in the “Results” section, but, a detailed explanation is required for each model.

Thanks for the comment. An explanation of the results for each model was added in line 152:

The results for the second, third, and fourth-order polynomial models clearly show that the coefficients of the higher-order terms than the quadratic are much less significant, which is why these coefficients' results are much lower than the coefficients of the terms quadratic, linear, and independent.

The exponential results model indicates similar exponents values. However, a different sign is shown in the coefficients to allow the change of slope direction with this model. This model shows a slowly rising slope and a slow descending slope, unlike polynomials. This is caused by the difference in the magnitude of the coefficients of both exponentials.

For the Gaussian function, the three parameters calculated are coefficient a, which is the value of the highest point of the bell, b is the position of the center of the bell, and c is the standard deviation. Thus, parameter a is determined by the magnitude of the measurements, and the average frequency of the measurements determines parameter b. Finally, parameter c reflects the variability of the model.

For the sinusoidal function, parameter a is the amplitude, which is why in magnitude, it is virtually the same as the parameter an of the Gaussian model. The angular velocity is coefficient b, and the phase is coefficient c

There is a significant lack in explaining the plots and the results presented in Table 3-4. For e.g., in Table 3 why the values of coefficients ‘b’ and ‘c’ are very high compare to the values of the coefficients for other models. A detailed significance of the results obtained from the analyses will provide a clear idea about performing the work to the readers along with the improvements.

Thanks for the comment, Table 3 and the different values for each coefficient was explained in the previous point. Table 4 and the plots were explained in line 187.

The Figures of the models show symmetric behavior for the quadratic, gaussian, and sinusoidal models. In contrast, the cubic, quartic and exponential models show asymmetric models with different rising and falling slopes. This effect can be observed more clearly in the Gaussian model (Figure [7]).

The value of the parameters was adjusted based on the RMSE. However,  this value does not reflect the data adjustment or the variability of the models. Therefore, the $R^2$ and the standard deviation of each model are in Table 4. It is observed that practically all the models share the same RMSE, the exception being the exponential model, which presents the most significant error. However, the third and fourth-order polynomial models show the lowest standard deviation and the cubic model the worst $R^2$. In general terms, the second-order, Gaussian, and sinusoidal models offer the best behavior practically the same in both RMSE, $R^2$, and standard deviation.

Overall, after a clear presentation of the results and restructured the manuscript, it has good potential for publication.

Thanks for all the comments.

Round 2

Reviewer 2 Report

The changes are satisfactory and I accept them. I recommend this paper for publication.

Reviewer 3 Report

The reviewer has been satisfied with the reply to the comments of the authors by incorporating them in the revised manuscript. The revised manuscript can be accepted for publication in its present form.